# Cardiac Exposure Related to Adjuvant Radiotherapy in Patients Affected by Thymoma: A Dosimetric Comparison of Photon or Proton Intensity-Modulated Therapy

**DOI:** 10.3390/cancers17020294

**Published:** 2025-01-17

**Authors:** Antonio Marco Marzo, Luca Cozzi, Davide Franceschini, Luca Dominici, Ruggero Spoto, Francesco Laurelli, Pasqualina Gallo, Lucia Paganini, Giacomo Reggiori, Federica Brilli, Alessandra Caracciolo, Ciro Franzese, Marco Francone, Marta Scorsetti

**Affiliations:** 1Radiotherapy and Radiosurgery Department, IRCCS Humanitas Research Hospital, Via Manzoni 56, 20089 Milan, Italy; antonio.marzo@humanitas.it (A.M.M.); luca.cozzi@humanitas.it (L.C.); luca.dominici@humanitas.it (L.D.); ruggero.spoto@humanitas.it (R.S.); pasqualina.gallo@humanitas.it (P.G.); lucia.paganini@humanitas.it (L.P.); giacomo.reggiori@humanitas.it (G.R.); ciro.franzese@hunimed.eu (C.F.); marta.scorsetti@hunimed.eu (M.S.); 2Department of Biomedical Sciences, Humanitas University, Via Rita Levi Montalcini 4, 20090 Milan, Italy; francesco.laurelli@st.hunimed.eu (F.L.); federica.brilli@humanitas.it (F.B.); alessandra.caracciolo@humanitas.it (A.C.); marco.francone@hunimed.eu (M.F.); 3Department of Diagnostic and Interventional Radiology, IRCCS Humanitas Research Hospital, Via Manzoni 56, 20089 Milan, Italy

**Keywords:** thymoma, thymic carcinoma, adjuvant radiotherapy, cardiac substructures, cardiac toxicity, proton therapy

## Abstract

Intensity-modulated proton therapy was compared to volumetric modulated arc therapy in this dosimetric study on patients who were candidates for adjuvant radiotherapy for thymic neoplasms. Fourteen cardiac substructures were identified, with the help of expert cardio-radiologists. Twenty nine patients were enrolled in the analysis. RT dose was 50 Gy in 25 fractions. The present study shows significantly better sparing of most of the cardiac substructures with proton therapy. The clinical impact of this dosimetric advantage should be tested in a clinical setting.

## 1. Introduction

Thymoma is a rare tumour representing only 0.2–1.5% of all malignancies, with an incidence of 0.13–0.32/100,000/year [1]. It arises from the epithelium of the thymus in the anterior mediastinum, and it typically occurs in adults aged between 40 and 70 years [2]. Although rare, it is the most frequent mediastinal tumour (20–50%) [3]. Surgery is the cornerstone of treatment, because the vast majority of thymomas are localized at presentation [4]. The Masaoka pathological staging and the completeness of resection are the most important predictors of outcome [5]. Post-operative radiation therapy is recommended after incomplete surgery or for advanced diseases (Masaoka stage III-IVa) [6], as stated in the current NCCN guidelines [7]. Recommended doses are 45–50 Gy for clear or close margins, 54–60 Gy for positive margins, and 60–70 Gy for patients with gross residual disease or in the definitive setting [7]. Prognosis is generally quite good, even for advanced diseases; the 5-year and 10-year overall survival are reported as 92% and 88% for stage I, 82% and 70% for stage II, 68% and 57% for stage III, and 61% and 38% for stage IV [8]. Given this excellent outcome, the majority of these patients will become long-term survivors and are thus at risk of developing late toxicities, such as radiation-induced heart disease (RIHD) [9]. Historically, the heart has been considered a whole organ, both in terms of contouring and dose constraints. More recently, research has focused on cardiac substructures (CSs), some of which are considered particularly radio-sensitive. Stam et al. for the first time reported an association between the dose received by the left atrium and superior vena cava and non-cancer-related death in a cohort of patients treated with SBRT for NSCLC [10]. In patients treated with adjuvant radiation therapy for breast cancer, Piroth et al. found a correlation between the risk of developing Major Adverse Cardiac Events (MACEs) and the specific dose parameters of the left ventricle and the left anterior descending coronary artery [11]. Similarly, a study conducted by Cai et al. evaluated the predictive value of cardiac substructure dosimetric parameters for cardiac toxicity after definitive radiation therapy in locally advanced oesophageal cancer [12]. The authors found that a model based on dose parameters of the left anterior descending coronary artery (LADCA) and the left circumflex artery (LCA) was associated with the prediction of severe acute coronary syndrome and/or congestive heart failure.

On the basis of these and other studies, van der Pol et al. extracted dose constraints for CSs and tested them in a population treated with MR-based SBRT for lung cancer, demonstrating that CSs can be successfully spared, without compromising tumour coverage nor increasing the dose to other OARs [13].

Given the location of the thymus in the anterior mediastinum, the radiation dose to the heart and its substructures is potentially high in patients undergoing radiotherapy for thymoma. A strategy to reduce it could be to use intensity-modulated proton beam therapy (IMPT).

Proton beams deliver most of their energy near the end of proton courses, a phenomenon known as the Bragg peak; when compared with megavoltage photon beams, they allow for low distant-to-target dose deposition and show a reduction in the integral dose with a low-dose bath [14].

Vogel et al. compared the dose received by OARs with IMRT versus IMPT in a cohort of 22 patients treated with adjuvant radiation therapy for thymic cancers and found a statistically significant reduction in the mean heart dose with IMPT, which translated to a lower predicted risk of late MACEs [15]. The purpose of this study is to investigate whether proton therapy compared to photon therapy could better spare cardiac substructures in a cohort of patients treated with adjuvant radiotherapy for thymoma, given prespecified dose constraints.

## 2. Material and Methods

This is a retrospective, single-institution, dosimetric study. Patients treated with adjuvant RT for radically resected thymoma from 2012 to 2022 were selected for the analysis. All patients underwent a contrast-enhanced simulation CT acquired in free-breathing conditions with a 3 mm slice thickness. The clinical target volume (CTV) included the thymic bed, the surgical clips and any potential area of residual disease. The planning target volume (PTV) was generated by an isotropic 5 mm expansion from the CTV. All plans were re-optimized to have a homogeneous dose prescription, 50 Gy in 25 daily fractions, normalised to the mean of the PTV. The following fourteen cardiac substructures (CSs) were considered for the dosimetric comparison: aortic valve (AO_Valve), ascending aorta (Asc_AO), right coronary artery (RCA), left circumflex artery (LCA), inferior vena cava (IVC), left atrium (LA), left anterior descending artery (LADCA), left ventricle (LV), pulmonary artery (PA), pericardium, pulmonary veins (PVs), right atrium (RA), right ventricle (RV) and superior vena cava (SVC). Every CS was manually contoured by a single radiation oncologist using a fixed window setting (WL: 40, WW: 400) following the guidelines of the following:Duane F et al. for coronary arteries and cardiac chambers [16];RTOG 1106 for pericardium and great vessels [17];Socha J et al. for the aortic valve [18].

All of these contours were revised by a team of radiation oncologists specialised in thoracic radiotherapy and radiologists specialised in cardiac imaging.

We generated new plans both with photon and proton therapy by giving selected dose constraints for some CSs (Table 1). Due to the rarity of thymoma, no constraints for cardiac substructures were previously determined in this setting. Therefore, we utilised proposed dose constraints extracted from the literature based on other, more common thoracic malignancies (Table 1).

The photon-based radiation therapy plans utilised were optimised in adherence to the volumetric modulated arc therapy (VMAT) technique, specifically its RapidArc variant, employing high-energy photon beams devoid of flattening filters characterised by a beam quality of 6 megavolts (MV) and planned for a TrueBeam linear accelerator (Varian Medical Systems, Palo Alto, Santa Clara, CA, USA). The process of optimization was conducted utilising the Eclipse treatment planning system, version 15 (Varian Medical Systems, Palo Alto, Santa Clara, CA, USA) and the final dose distribution was recalculated using the Acuros-XB type-c algorithm.

The proton-based radiation therapy plans utilised were optimised with the ProBeam proton system (Varian Medical Systems, Palo Alto, Santa Clara, CA, USA) used as a source of beam data for the optimization and calculation of intensity-modulated proton therapy (IMPT) plans with the beam spot scanning technique. The dose distribution optimization was performed in Eclipse using the Nonlinear Universal Proton Optimiser (NUPO, v16.0), while for the final dose calculation, the Proton Convolution Superposition algorithm (v16.0) was applied using a grid size of 2.5 mm, without constant relative biological effectiveness (RBE).

### Statistical Analysis

Non-parametric tests have been used to compare the DVH of the VMAT versus IMPT plans. Numerical analysis has been carried on the various organs at risk and reported in terms of appropriate dose–volume metrics. We reported, following ICRU recommendations, the mean dose (D_mean_), the volume receiving at least 5 Gy (V_5Gy_) and the near-to-max (D_near-to-max_ with appropriate volume thresholds) of every cardiac substructure. The advantage of sparing cardiac substructures with IMPT was considered statistically significant if the *p*-values were less than 0.05.

## 3. Results

Twenty-nine patients were analysed. Clinical indication for adjuvant RT was mainly due to disease stage or resection status. Most patients were affected by thymoma (24/29, 82.7%), stage III according to Masaoka Koga (21/29, 72.4%). Five patients (17.3%) had pN-positive disease. All but five patients underwent R0 resection. No R2-resected patients was included in this analysis. Patients were treated with different doses according to disease stage and resection status, as per NCCN guidelines. However, to allow for a homogeneous comparison between photon and proton plans, all patients were re-optimized with a dose of 50 Gy in 25 fractions.

New VMAT and IMPT plans were compared based on doses delivered to cardiac substructures. In general, we found that IMPT can better spare cardiac substructures in terms of D_near-to-max_ dose to ≤0.035 cm^3^ (Table 2), D_mean_ (Table 3) and V_5Gy_ (Table 4).

Specifically, RCA D_near-to-max_ and D_mean_ were reduced by about 5 Gy and 3 Gy, respectively (*p* = 0.000). A larger advantage was detected for AO_Valve. Indeed, D_near-to-max_ and D_mean_ were reduced by approximately 6 Gy, and V_5Gy_ was reduced from 56% to 6% (*p =* 0.000). Even LADCA was also better spared with IMPT in terms of D_near-to-max_ (18.1 ± 19.5 Gy vs. 26.3 ± 17.7 Gy, *p* = 0.000), D_mean_ (8.3 ± 12 Gy vs. 12.6 ± 12.4 Gy, *p* = 0.000) and V_5Gy_ (26.6 ± 34.1% vs. 55.1 ± 26.6%, *p* = 0.002). Figure 1 shows the dose–volume histograms of IMPT and VMAT plans for some selected cardiac substructures.

We also evaluated how frequently IMPT and VMAT plans were able to respect dose constraints for cardiac substructures. We found that these objectives were generally easier to reach with IMPT (Table 5).

We also noticed that for some cardiac substructures like AO_Valve, the left and right atrium, the LADCA and the ascending aorta, even IMPT plans were not able to reach the plan’s objectives in most patients due to the challenging position of target volumes.

## 4. Discussion

We report a dosimetric comparison in terms of cardiac substructure sparing among patients treated with adjuvant radiotherapy for thymoma. Indeed, there is a growing interest nowadays in radiation-induced cardiac disease, especially in thoracic malignancies.

Cardiac toxicity from radiotherapy has been observed initially in patients treated for breast cancer and Hodgkin lymphoma [24,25]. Darby et al. reported for the first time in breast cancer survivors the existence of a linear relationship between the risk of MACEs and the dose received by the heart: for every Gy increase in the mean heart dose, the risk increased by 7.4% [24]. Among patients affected by Hodgkin lymphoma, an increased risk of cardiovascular disease is detectable up to 40 years after radiation treatment [25]. Radiation exposure of the heart was associated with coronary heart disease and valvular heart disease [26], and the risk of cardiac events was higher in patients with cardiovascular risk factors or pre-existing cardiac disease. The incidence of cardiac events also increases after radiotherapy for oesophageal cancer, and a mean heart dose > 15 Gy has been recognized as a predictor for cardiac toxicity [27]. RTOG 0617 was the first trial demonstrating the relevance of cardiac toxicity in lung cancer [28].

This was a phase III trial comparing dose-escalated radiotherapy to standard-dose radiotherapy in patients affected by locally advanced, unresectable lung cancer. Overall survival was shorter in the experimental arm, and it was linked to the higher dose received by the heart in terms of V_40Gy_ (the volume of the whole heart receiving at least 40 Gy) [29]. More recently, among this population, McWilliam et al. have identified, through a voxel-based analysis, a region called the “base of heart” which received the highest dose in the dose-escalated arm, thus correlating with the excessive mortality reported [30]. In a cohort of patients treated with adjuvant RT for thymoma, cardiovascular disease was the leading non-malignant cause of death. The mean heart dose was found to be an independent risk factor in this case [31].

In all of these studies, the heart has been considered a whole organ, both in terms of contouring and dose constraints. However, more recently, a variety of cardiac substructures have been identified and related to a particular clinical event. Indeed, in our analysis, we directly compared the newly optimised photon-based plans with a proton-based plan in terms of the dose received by the previously identified cardiac substructures. In general, we found that IMPT was better than VMAT in sparing cardiac substructures.

We particularly focused our attention on some cardiac substructures already related to clinical heart toxicity in the literature. Moving from a sub-analysis of RTOG 0617, the group of Manchester researchers identified a region called the base of the heart which is strongly associated with worse survival for cardiac events [19]. The base of the heart is formed posteriorly by the left atrium and the pulmonary veins, and anteriorly by the right ventricular outflow tract, the aortic root, the origin of the coronary arteries and the junction between the superior vena cava and the right atrium, where the sino-atrial node is located. McWilliam et al. identified that the maximum dose to the combined cardiac region encompassing the right atrium, the right coronary artery and the ascending aorta with the aortic valve was linked with patient survival, and they finally recommended that the D_near-to-max_ dose to this combined region should be kept below 23 Gy. In our analysis, with few exceptions, we found that IMPT could reduce the dose to all of the cardiac substructures forming the base of the heart. Piroth et al. [11] published a review in which they analysed all of the studies that reported clinically late cardiac events occurring after adjuvant radiotherapy for breast cancer in order to find a dose–effect relationship either for the mean heart dose or for doses to cardiac substructures. They recommend that breast cancer radiotherapy planning should include constraints not only for the whole heart but also for the left ventricle and LADCA. In particular, the mean dose to the LV should be kept under 3 Gy and the percentage of the left ventricle receiving 5 Gy (V_5Gy_) should be less than 17% in order to ensure adequate heart protection. In our study, these two dose constraints for the LV have been adhered to more often by IMPT rather than VMAT plans. Moreover, they recommend that the mean dose to the LADCA should be less than 10 Gy. Although we did not meet that constraint, since this is a different clinical scenario, we found a general and statistically significant reduction in the mean dose to the LADCA using IMPT. Still focusing on the LADCA, Zureick et al. retrospectively analysed 375 consecutive patients treated with adjuvant radiotherapy for left-sided breast cancer with the aim of investigating whether dose to the LADCA correlated with cardiac events [20]. They found that 36 patients experienced a cardiac event, and that, after multivariate analysis, the maximum dose to the LADCA was associated with an increased risk of cardiac events, in particular when it was above 6.7 Gy. Also, in this case, we did not meet those constraints either by using IMPT or VMAT plans, but we found a statistically significant reduction in Dnear-to-max for the LADCA using proton-based radiotherapy. Evaluating our data, it should be kept in mind that, in thymoma patients, the clinical target volume often includes the anterior pericardium, thus increasing the dose to the anterior cardiac substructures, such as the LADCA. Cai et al. analysed 716 patients treated with definitive radiation therapy for stage II-III oesophageal cancer in order to investigate the predictive value of cardiac substructure dosimetric parameters for cardiac toxicity [12]. Sixty-eight (15.7%) developed grade ≥ 3 cardiac toxicities and they found that models based on dosimetry of the left circumflex artery showed a favourable predictive accuracy for grade ≥ 3 cardiac events, thus recommending that radiation dose to the LCA should be monitored to help reduce the occurrence of heart toxicity. We can affirm that in our cohort of patients, the left circumflex artery was better spared by IMPT in terms of D_near-to-max_, D_mean_ and V_5Gy_ (Figure 1F).

We want to highlight how the cardiotoxicity mechanisms of RT are still unclear. Among the various effects that radiation can have on the heart, microvascular damage and sustained inflammation are probably the most relevant [21]. Since no safe dose limits have been found, any attempt to minimize cardiac exposure is clinically relevant. In this setting, IMPT is one of the possible improvements.

Our study has several limitations. First, it is a purely dosimetric analysis, so we did not follow our patients to assess their clinical outcome. Therefore, we lack clinical endpoints correlating with the dosimetric parameters that we found. Secondly, as there are no specific dose constraints for cardiac substructures in the thymoma setting, we applied dose constraints derived from other and more frequent cancer settings, in particular breast, lung and oesophageal cancer. The position of the clinical target volume in these cases is very different from the case of thymoma, where the target is nearer to the heart and often comprises the pericardium and part of the great vessels. Therefore, we found some difficulties in adhering to the dose constraints, especially for the most superior and anterior cardiac substructures.

Technically, patients were treated with a free-breathing approach. It is likely that, using a gated approach, at least some patients could have derived a major benefit in terms of cardiac sparing, especially with deep inspiration breath holding.

Another possible limitation is the low number of analysed patients. This is partially due to the rarity of the disease and also due to the time-consuming work needed for re-contouring, re-planning and re-optimization for both the photon and proton plans. However, since this is a purely dosimetric study, even this low number is sufficient to prove, in our view, the possible theoretical advantage of PT in the treatment of these patients.

## 5. Conclusions

We demonstrated that cardiac substructures can be successfully spared in intensity-modulated adjuvant proton therapy for thymoma in the majority of our patient cohort. Future prospective, clinical studies are needed to establish a possible relationship between dose parameters of cardiac substructures and the development of specific cardiac events.

## Figures and Tables

**Figure 1 cancers-17-00294-f001:**
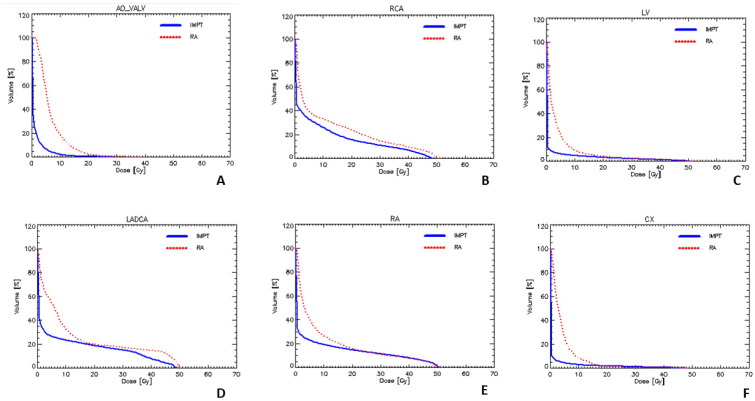
Dose–volume histogram (DVH) of aortic valve (**A**), right coronary artery (**B**), left ventricle (**C**), left anterior descending coronary artery (**D**), right atrium (**E**) and left circumflex artery (**F**) with proton (full line) and photon (dashed line) plans.

**Table 1 cancers-17-00294-t001:** Proposed dose constraints for cardiac substructures.

Cardiac Substructure	Dose Constraints (EQD2)	Setting
Ascending Aorta [10]	D80% < 0.41 Gy	Lung SBRT
Right Coronary Artery [19]	D_0.03cc_ < 23 Gy	Lung CFRT
Left Circumflex Artery [20]	V15 < 14%	Breast Cancer
Left Atrium [10]	D_0.03cc_ < 6.5 Gy	Lung SBRT
Left Anterior Descending Coronary Artery [11,20,21]	V15 < 10% V30 < 2% V40 < 1% D_mean_ < 2.8 Gy D_0.03cc_ < 6.7 Gy	Breast Cancer
Left Ventricle [11,20]	D_mean_ < 3 Gy V5 < 17% V15 < 1%	Breast Cancer
Pulmonary Artery [22]	V40 < 80% V45 < 68% V50 < 45% V55 < 32%	Lung CFRT
Right Atrium [23]	D_0.03cc_ < 19.1 Gy	Lung CFRT
Right Ventricle [10]	D10% < 0.7 Gy	Lung SBRT
Superior Vena Cava [10]	D90% < 0.59 Gy	Lung SBRT

**Table 2 cancers-17-00294-t002:** Mean D near-to-max (dose to ≤0.035 cm^3^) value in proton and photon plans (in bold, statistically significant values).

Structure	Mean Value VMATnew	Mean Value IMPT	*p* (Student)	*p* (Fisher’s Sign)
**Aortic Valve**	**15.14 ± 11.75 Gy**	**8.86 ± 8.57 Gy**	**0.000**	**0.000**
**Ascending Aorta**	**51.24 ± 0.56 Gy**	**52.19 ± 0.73 Gy**	**0.000**	**0.000**
**Right Coronary Artery**	**27.76 ± 17.57 Gy**	**22.14 ± 15.59 Gy**	**0.000**	**0.000**
**Left Circumflex Artery**	**11.89 ± 9.71 Gy**	**5.73 ± 11.56 Gy**	**0.000**	**0.000**
**Inferior Vena Cava**	**3.71 ± 4.32 Gy**	**1.25 ± 4.91 Gy**	**0.006**	**0.000**
**Left Atrium**	**23.08 ± 16.86 Gy**	**18.50 ± 17.91 Gy**	**0.001**	**0.001**
**Left Anterior Descending Coronary Artery**	**26.30 ± 17.73 Gy**	**18.10 ± 19.52 Gy**	**0.000**	**0.000**
Left Ventricle	23.60 ± 18.72 **Gy**	19.52 ± 22.75 **Gy**	0.015	0.132
**Pulmonary Artery**	**51.42 ± 1.04 Gy**	**52.25 ± 0.92 Gy**	**0.000**	**0.000**
Pulmonary Veins	39.71 ± 13.72 **Gy**	39.34 ± 15.84 **Gy**	0.603	0.229
Right Atrium	37.28 ± 18.35 **Gy**	34.97 ± 18.45 **Gy**	0.020	0.132
Right Ventricle	33.39 ± 19.87 **Gy**	35.31 ± 20.43 **Gy**	0.096	0.004
Superior Vena Cava	47.97 ± 6.85 **Gy**	47.87 ± 10.10 **Gy**	0.902	0.012

**Table 3 cancers-17-00294-t003:** Mean D_mean_ values in proton and photon plans (in bold, statistically significant values).

Structure	Mean Value VMATnew	Mean Value IMPT	*p* (Student)	*p* (Fisher’s Sign)
**Aortic Valve**	**6.97 ± 4.07 Gy**	**1.28 ± 1.65 Gy**	**0.000**	**0.000**
**Ascending Aorta**	**29.26 ± 6.17 Gy**	**25.35 ± 6.95 Gy**	**0.000**	**0.000**
**Right Coronary Artery**	**11.42 ± 11.0 Gy**	**8.06 ± 10.22 Gy**	**0.000**	**0.000**
**Left Circumflex Artery**	**4.36 ± 2.70 Gy**	**1.01 ± 2.90 Gy**	**0.000**	**0.000**
**Inferior Vena Cava**	**1.70 ± 1.79 Gy**	**0.08 ± 0.37 Gy**	**0.000**	**0.000**
**Left Atrium**	**5.79 ± 3.35 Gy**	**1.14 ± 2.38 Gy**	**0.000**	**0.000**
**Left Anterior Descending Coronary Artery**	**12.60 ± 12.3 Gy**	**8.29 ± 12.01 Gy**	**0.000**	**0.000**
**Left Ventricle**	**4.38 ± 4.69 Gy**	**1.64 ± 3.14 Gy**	**0.000**	**0.000**
**Pulmonary Artery**	**28.31 ± 7.04 Gy**	**23.09 ± 9.54 Gy**	**0.000**	**0.000**
**Pulmonary Veins**	**12.02 ± 6.38 Gy**	**6.31 ± 5.65 Gy**	**0.000**	**0.000**
**Right Atrium**	**9.57 ± 8.40 Gy**	**6.93 ± 8.76 Gy**	**0.000**	**0.000**
**Right Ventricle**	**6.4 ± 7.00 Gy**	**3.92 ± 5.13 Gy**	**0.000**	**0.000**
Superior Vena Cava	29.75 ± 11.1 **Gy**	25.27 ± 16.60 **Gy**	0.003	0.132

**Table 4 cancers-17-00294-t004:** Mean V_5Gy_ values in proton and photon plans (in bold, statistically significant values).

Structure	Mean Value (%) VMATnew	Mean Value (%) IMPT	*p* (Student)	*p* (Fisher’s Sign)
**Aortic Valve**	**56.42 ± 41.71**	**6.96 ± 12.30**	**0.000**	**0.000**
**Ascending Aorta**	**92.02 ± 9.99**	**71.36 ± 13.40**	**0.000**	**0.000**
**Right Coronary Artery**	**39.43 ± 35.97**	**33.06 ± 30.70**	**0.061**	**0.054**
**Left Circumflex Artery**	**26.67 ± 19.58**	**4.46 ± 10.27**	**0.000**	**0.000**
Inferior Vena Cava	6.58 ± 18.63	0.46 ± 2.44	0.088	0.035
**Left Atrium**	**37.59 ± 25.88**	**4.42 ± 8.40**	**0.000**	**0.000**
**Left Anterior Descending Coronary Artery**	**55.11 ± 26.67**	**26.58 ± 34.13**	**0.000**	**0.002**
**Left Ventricle**	**22.34 ± 25.91**	**6.31 ± 11.25**	**0.000**	**0.000**
**Pulmonary Artery**	**97.02 ± 5.73**	**66.33 ± 18.45**	**0.000**	**0.000**
**Pulmonary Veins**	**61.33 ± 26.00**	**22.30 ± 14.79**	**0.000**	**0.000**
**Right Atrium**	**40.13 ± 34.96**	**22.87 ± 25.54**	**0.000**	**0.000**
Right Ventricle	25.95 ± 30.20	14.89 ± 17.68	0.001	0.084
**Superior Vena Cava**	**94.78 ± 8.62**	**67.91 ± 32.39**	**0.000**	**0.000**

**Table 5 cancers-17-00294-t005:** Percentage of patients achieving proposed dose constraints in proton and photon plans.

STRUCTURE	CONSTRAINT	IMPT	VMATnew
Aortic Valve	D_mean_ < 1.62 Gy	23\29 (79.3%)	0\29 (0%)
Aortic Valve	V_5Gy_ = 0%	9\29 (31%)	5\29 (17.2%)
Ascending Aorta	D80% < 0.41 Gy	1\29 (3.4%)	0\29 (0%)
Right Coronary Artery	D_0.03cc_ < 23 Gy	18\29 (62%)	11\29 (37%)
Left Circumflex Artery	D_mean_ < 23.7 Gy	29\29 (100%)	28\29 (96.5%)
Left Circumflex Artery	V_15Gy_ < 14%	26\29 (89.6%)	22/29 (75.8%)
LADCA	D_mean_ < 2.8 Gy	5\29 (17.2%)	0\29 (0%)
LADCA	D_0.03cc_ < 6.7 Gy	8\29 (27.5%)	0\29 (0%)
LADCA	V_15Gy_ < 10%	17\29 (58.6%)	13\29 (44.8%)
LADCA	V_30Gy_ < 2%	19\29 (65.5%)	18\29 (62%)
LADCA	V_40Gy_ < 1%	19\29 (65.5%)	19\29 (65.5%)
Left Atrium	D_0.03cc_ < 6.5 Gy	3\29 (10.3%)	0\29 (0%)
Left Ventricle	D_mean_ < 3 Gy	25\29 (86.2%)	15\29 (51.7%)
Left Ventricle	V_5Gy_ < 17%	25\29 (86.2%)	15\29 (51.7%)
Left Ventricle	V_15Gy_ < 1%	18\29 (62%)	14\29 (48.2%)
Right Atrium	D_0.03cc_ < 19.1 Gy	5\29 (17.2%)	6\29 (20.6%)
Right Ventricle	D10% < 0.7 Gy	9/29 (31%)	0/29 (0%)
Pulmonary Artery	V_40Gy_ < 80%	29/29 (100%)	29/29 (100%)
Pulmonary Artery	V_45Gy_ < 68%	29/29 (100%)	29/29 (100%)
Pulmonary Artery	V_50Gy_ < 45%	29/29 (100%)	29/29 (100%)
SVC	D90% < 0.59 Gy	13/29 (44.8%)	0/29 (0%)

## Data Availability

The research data are stored in an institutional repository and will be shared upon request to the corresponding author.

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
