# Peer review of "Cardiac Exposure Related to Adjuvant Radiotherapy in Patients Affected by Thymoma: A Dosimetric Comparison of Photon or Proton Intensity-Modulated Therapy"

_cancers, 2025, doi:10.3390/cancers17020294_

Round 1
Reviewer 1 Report
Comments and Suggestions for Authors
Currently the appropriate control arm might be respiratory gating or DIBH which might further improve IMRT based treatment dosimetry, [still recognizing the proximity of the tx volume will by necessity always include some cardiac substructure dosing]. This could be commented in the discussion. The point is that VMAT with free breathing might be improved upon with a change to gating, without resorting to $$ protons.
21 "lodge" is not an English medical term
23 PBT is not spelled out
23 "proton therapy" is inconsistent with term used in title "proton intensity modulated therapy"
25-28 most of this should be in methods and not in abstract
31 "impt" is not spelled out
59 "historically in literature", suggest "historically" or "literature" but not both...redundant
80 "lodge" is not an English medical term
85 replace semi colon with period
67 & 91 MACE should be shown in line 61 where it is spelled out, and abbreviated after
162 D near-to-max....if this is D2% then it should be specified.
164-5 Table 2 has no units
168-9 Table 3 has no units
Table 5 formating/justification is irregular in columns 4 & 5
245 "Piroth" has no reference number given, although previously mentioned
Author Response
Thanks for your comments
Currently the appropriate control arm might be respiratory gating or DIBH which might further improve IMRT based treatment dosimetry, [still recognizing the proximity of the tx volume will by necessity always include some cardiac substructure dosing]. This could be commented in the discussion. The point is that VMAT with free breathing might be improved upon with a change to gating, without resorting to protons.
Correct, we added a comment on that in the discussion, where acknowledging the limitations of the present study
21 "lodge" is not an English medical term
Changed
23 PBT is not spelled out
Spelled
23 "proton therapy" is inconsistent with term used in title "proton intensity modulated therapy"
We chose to use IMPT and changed it in the manuscript
25-28 most of this should be in methods and not in abstract
We preferred to give also some practical information on the methodology followed for this study even in the abstract section
31 "impt" is not spelled out
Spelled
59 "historically in literature", suggest "historically" or "literature" but not both...redundant
Changed
80 "lodge" is not an English medical term
Changed
85 replace semi colon with period
Done
67 & 91 MACE should be shown in line 61 where it is spelled out, and abbreviated after
Done
162 D near-to-max....if this is D2% then it should be specified.
It is dose to ≤0.035 cm3, now specificied in the manuscript
164-5 Table 2 has no units
Added
168-9 Table 3 has no units
Added
Table 5 formating/justification is irregular in columns 4 & 5
Modified
245 "Piroth" has no reference number given, although previously mentioned
Added, thanks
Reviewer 2 Report
Comments and Suggestions for Authors
Please read your paper carefully before submitting it and correct typos, abbreviations, spacing, line breaks, etc. Even if the research is original and the results are creative, it is not a good paper if the paper is full of errors such as typos, spacing, abbreviations, and line breaks.
Table 1.
D80%<0, 41Gy. D0.03cc<6, 7Gy. Dmean<2, 8Gy
What are these?
This study is a dosimetric study comparing the cardiac doses of IMRT with IMPT in thymoma patients who underwent postop RT. However, in the discussion section, the authors focus on explaining the correlation between exposed radiation doses and the occurrence of cardiac toxicities. Authors should write in the Discussion section what is directly relevant to the results of this study.
A number of dosimetric studies have already been published comparing the cardiac doses of IMRT and IMPT. I don't think readers will be able to get new and original information from this paper.
Author Response
Thanks for your comment
Please read your paper carefully before submitting it and correct typos, abbreviations, spacing, line breaks, etc. Even if the research is original and the results are creative, it is not a good paper if the paper is full of errors such as typos, spacing, abbreviations, and line breaks.
Thanks for your observation
Table 1.
D80%<0, 41Gy. D0.03cc<6, 7Gy. Dmean<2, 8Gy
What are these?
These are possibile dose constraints for cardiac substructures derived from literature along with the clinical setting in which they have been proposed or used
This study is a dosimetric study comparing the cardiac doses of IMRT with IMPT in thymoma patients who underwent postop RT. However, in the discussion section, the authors focus on explaining the correlation between exposed radiation doses and the occurrence of cardiac toxicities. Authors should write in the Discussion section what is directly relevant to the results of this study.
We tried to do this at the best of our capabilities. We focused on correlation between exposed radiation doses and the occurrence of cardiac toxicities since our is a dosimetric and not clinical study. This correlation can ben clinically relevant if clinical events have been already found in literature, in our opinion
A number of dosimetric studies have already been published comparing the cardiac doses of IMRT and IMPT. I don't think readers will be able to get new and original information from this paper.
Reviewer 3 Report
Comments and Suggestions for Authors
The authors present a series based on a single-centre experience of a dosimetric comparison of intensity modulated photons or protons on cardiac damage related to adjuvant radiotherapy in patients with thymoma. the topic is original and interesting, although the number of patients included is limited.
the main limitation of the study is the lack of information on patient follow-up.
I have a few suggestions for the authors:
- In line 23, the acronym PBT should be described.
- NCCN guidelines recommend adjuvant radiotherapy for locally advanced disease: change the text (line 49).
- More details on histopathological diagnosis according to the WHO classification and Masaoka-Koga staging should be given in a table.
- The process of radiation-induced heart disease could be described in the introduction or discussion, also to explain the different effects of proton and photon therapy on cardiac structures.
Author Response
Thanks for your comments
The authors present a series based on a single-centre experience of a dosimetric comparison of intensity modulated photons or protons on cardiac damage related to adjuvant radiotherapy in patients with thymoma. the topic is original and interesting, although the number of patients included is limited.
the main limitation of the study is the lack of information on patient follow-up.
I agree with you, however this was not the intent of the present study, since this is a purely dosimetric study as declared and acknowledged
I have a few suggestions for the authors:
- In line 23, the acronym PBT should be described.
Described
- NCCN guidelines recommend adjuvant radiotherapy for locally advanced disease: change the text (line 49).
Changed
- More details on histopathological diagnosis according to the WHO classification and Masaoka-Koga staging should be given in a table.
Clinical informations have been listed in the first paragraph of results to avoid adding another table, since already 5 tables are present. Moreover, the focus of the study is not the clinical indication neither the clinical outcome
- The process of radiation-induced heart disease could be described in the introduction or discussion, also to explain the different effects of proton and photon therapy on cardiac structures
Added some clarification on it and a reference
Reviewer 4 Report
Comments and Suggestions for Authors
This is a very thorough study both in its execution and its presentation.
Thymic "lodge" is an unusual term, not in widespread use - perhaps this needs to be described differently?
Author Response
Thanks for your comment
Thymic "lodge" is an unusual term, not in widespread use - perhaps this needs to be described differently?
Changed, thanks
Round 2
Reviewer 2 Report
Comments and Suggestions for Authors
1. Table 1. D80%<0, 41Gy. D0.03cc<6, 7Gy. Dmean<2, 8Gy
Do you mean D80%<0.41Gy. D0.03cc<6.7Gy. Dmean<2.8Gy? It seems there may be a typographical error in distinguishing between commas and periods. Please double-check and correct these values for clarity.
2. Some of the points from my earlier review may not have been addressed in the revision. Could you please ensure all comments are thoroughly reviewed and incorporated where appropriate?
Author Response
1. Table 1. D80%<0, 41Gy. D0.03cc<6, 7Gy. Dmean<2, 8Gy
Do you mean D80%<0.41Gy. D0.03cc<6.7Gy. Dmean<2.8Gy? It seems there may be a typographical error in distinguishing between commas and periods. Please double-check and correct these values for clarity.
That's correct, you can change commas with periods.
2. Some of the points from my earlier review may not have been addressed in the revision. Could you please ensure all comments are thoroughly reviewed and incorporated where appropriate?
I understand the reviewers comments. However, these suggestions were not incorporated in the discussion, since as previosuly stated, we focused on correlation between exposed radiation doses and the occurrence of cardiac toxicities since our is a dosimetric and not clinical study. This correlation can ben clinically relevant if clinical events have been already found in literature, in our opinion